# VolPy: Automated and scalable analysis pipelines for voltage imaging datasets

Changjia Cai[1], Johannes Friedrich[2], Amrita Singh[3], M. Hossein Eybposh[1], Eftychios A. Pnevmatikakis[2], Kaspar Podgorski[3]*, Andrea Giovannucci[1,4]*

1 Joint Department of Biomedical Engineering at University of North Carolina at Chapel Hill and North Carolina State University, Chapel Hill, North Carolina, United States of America, 2 Flatiron Institute, Simons Foundation, New York, New York, United States of America, 3 Janelia Research Campus, Howard Hughes Medical Institute, Ashburn, Virginia, United States of America, 4 Neuroscience Center, University of North Carolina at Chapel Hill, Chapel Hill, North Carolina, United States of America

* podgorskik@janelia.hhmi.org (KP); agiovann@email.unc.edu (AG)

**Data Availability Statement:** The code and datasets are available at the dev branch of the repository https://github.com/flatironinstitute/CalmAn and at the Zenodo repository https://

## Abstract

Voltage imaging enables monitoring neural activity at sub-millisecond and sub-cellular scale, unlocking the study of subthreshold activity, synchrony, and network dynamics with unprecedented spatio-temporal resolution. However, high data rates (>800MB/s) and low signal-to-noise ratios create bottlenecks for analyzing such datasets. Here we present *VolPy*, an automated and scalable pipeline to pre-process voltage imaging datasets. *VolPy* features motion correction, memory mapping, automated segmentation, denoising and spike extraction, all built on a highly parallelizable, modular, and extensible framework optimized for memory and speed. To aid automated segmentation, we introduce a corpus of 24 manually annotated datasets from different preparations, brain areas and voltage indicators. We benchmark *VolPy* against ground truth segmentation, simulations and electrophysiology recordings, and we compare its performance with existing algorithms in detecting spikes. Our results indicate that *VolPy*'s performance in spike extraction and scalability are state-of-the-art.

## Author summary

Roughly 290 million electrical action potentials occur every second in the human brain, facilitating the propagation of signals among cells in the nervous system and driving most of our daily operations. New methods in brain imaging are emerging that have the speed and resolution to capture events in the brain at the pace at which neurons typically communicate. These methods measure voltage in neurons by using light, and therefore can access very detailed brain signaling patterns. However, the adoption of these methods by a larger community, and not a restricted set of experts, is limited by the lack of computational tools, thereby greatly hindering progress in this field. In this paper, we present *VolPy*, a software framework that greatly facilitates the preprocessing of this new type of imaging datasets. This pipeline incorporates efficient and optimized algorithms that can identify neurons and extract their activity with great accuracy. The presented software will make this new imaging modality accessible to a wide audience.

zenodo.org/record/4515768/export/hx#.
YEf4K2RKgwQ. Data used in figures are included in
S1 Data. Mask R-CNN is available (only required
for retraining the network), from https://github.
com/matterport/Mask_RCNN. Mask R-CNN was
trained with the following tools: python 3.7.3,
tensorflow-gpu 1.14.0.

**Funding:** KP, AS are supported by the Howard
Hughes Medical Institute (salary and laboratory
resources), which had a role in data collection,
preliminary analysis, and preparation of the
manuscript. AG, CC are supported by the Arnold
and Mabel Beckman Foundation (summer salary of
AG and salary of CC) and the Kavli Foundation
(salary of CC to build graphical user interface that
was extended for the purpose of the current
paper). BF had a role in study design, decision to
publish, preparation of manuscript, data annotation
and analyses, data and code sharing infrastructure.
KF had a role in code sharing infrastructure.

**Competing interests:** The authors have declared
that no competing interests exist.

This is a *PLOS Computational Biology* Methods paper.

## Introduction

Recording subthreshold voltage changes and exact spike times in populations of neurons is necessary to dissect the details of information processing in the brain. Voltage imaging is currently the only technique that promises to achieve this goal with high spatio-temporal resolution. While methods have been developed to process voltage imaging data at mesoscopic scale and multi-unit resolution [1–3], to date there is no established pipeline for large-scale cellular-resolution analyses.

Voltage indicators have only recently become sensitive enough to generate large-scale recordings at cellular resolution [4–10]. Existing voltage imaging analysis pipelines have been developed for specific datasets, feature limited scalability and are difficult to use for the novice programmer [5, 6, 11, 12]. For instance, [5] introduces an iterative method named SpikePursuit, which detects lower-amplitude spikes using filters generated from easily-detected high-amplitude spikes. However, this method requires manual selection of neurons and is not optimized for parallel processing—i.e. distributing the workload among several CPUs to speed up computing—or scalability. Other techniques [6, 11, 12] rely on penalized matrix decomposition to facilitate denoising of fluorescence activity and utilize localNMF to perform an initial segmentation of neurons. However, these frameworks do not offer an adaptive and automated mechanism for spike extraction and are not integrated into a scalable and multi-platform framework. Further, while all these methods have been validated with simulated data and specific electrophysiological ground truth datasets, the lack of a unified benchmark has hindered the validation and comparison of all these approaches. This highlights the need for a validated and scalable pipeline for the automatic analysis of voltage imaging data, ideally embedded into a reusable and well documented format, an important requirement for broad community use.

Common techniques for the analysis of calcium imaging [13] data, a comparable recording modality, have not been systematically tested on voltage imaging datasets and their effectiveness on this new imaging modality is unclear. While signals recorded from calcium imaging are slow and present in the whole cytosol, voltage sensors are mostly expressed on the membrane and produce much faster signals. Therefore, while calcium imaging benefits from averaging in space and time, voltage imaging in general features lower SNR. Further, assumptions made by these methods about underlying signals may be violated in voltage imaging recordings. For instance, it might not be compatible with the matrix factorization techniques for calcium imaging that typically use a mean square error loss term [14–17]: (i) voltage imaging signals contribute less variance to recorded videos; (ii) voltage signals display both positive and negative fluctuations (excluding the methods [14, 17], which can handle this case); (iii) significant multiplicative noise may arise from light absorption in one-photon voltage recordings, which is not compatible with the mean square error loss term; (iv) signals are much faster, noisier, and with different underlying dynamics, not captured by biophysical models based on calcium imaging [18].

To address these shortcomings, we developed a new analysis pipeline for preprocessing voltage imaging data, called *VolPy*. The pipeline provides algorithms and routines to correct for motion artifacts, to automatically identify and segment neurons, to denoise voltage signals, and to extract action potentials and subthreshold signals. In this pipeline automatic detection and segmentation of neurons is performed by a convolutional neural network based on Mask R-CNN [19], which we trained using a corpus of manually segmented datasets. Spike extraction and signal denoising is performed using a more scalable and efficient version of the

SpikePursuit [5] algorithm, which we equipped with automatic initialization and extraction of subthreshold signals. Crucially, we integrated our pipeline within the *CaImAn* ecosystem [16], a popular suite of tools for single cell resolution brain imaging analysis already adopted by the neuroscience community.

We quantitatively evaluated the performance of *VolPy* on neuron segmentation and compared its performance against humans. We also compared *VolPy* with other algorithms on simulated and real datasets, as well as on simultaneous electrophysiology and voltage imaging data. Our results show that *VolPy* outperforms existing methods both in spike detection and scalability.

## Materials and methods

### Creation of a corpus of annotated datasets

To date there are no established annotated datasets for single cell localization and/or segmentation in cellular-resolution voltage imaging. Towards filling this gap, and with the goal of developing new supervised algorithms, we generated a corpus of 24 manually segmented datasets (Ground truth, GT) by combining annotations from three independent human labelers. To provide annotations, human labelers relied upon two summary images (mean and local correlation images, see Fig 1B and 1C, S1 Fig), which were built as follows:

**Mean image**: We averaged the movie across time for each pixel yielding a 2D image. We normalized the 2D image by subtracting the mean of its pixels and dividing by the standard deviation of its pixels. The normalization step enables different datasets to share the same scale as the input to the segmentation step.

**Correlation image**: The correlation image is a variation of that implemented in [20]. After removing the baseline of the movie by high-pass filtering, we averaged the temporal correlation of each pixel with its eight neighbor pixels yielding another 2D image. The resulting image was then mean-subtracted and divided by its standard deviation.

Guided by these visual cues, three annotators marked the contours of neurons using the FIJI ImageJ Cell Magic Wand tool plugin [21] (S1 Fig). Labelers were trained using a test dataset and instructed to look for ring or circle-shaped structures which were clear on either the mean or the correlation image. An exception to this rule were blood vessels perpendicular to the imaging plane, which looked like dark circles in the mean image and bright circles in the correlation image. We then generated a consensus ground truth by combining the three annotations. For a neuron to be included in the consensus ground truth, it either had to be selected by two or more annotators, or all annotators had to agree on accepting it in a separate follow-up session. The finally selected pixels associated to a consensus mask was selected based on masks provided by the most experienced annotators of the three. Summary information about the annotated datasets is reported in Table 1.

### A novel analysis pipeline for voltage imaging

We propose a novel scalable pipeline for automated analysis that performs the preprocessing steps required to extract spikes and subthreshold activity from voltage imaging movies (Fig 1). First, input data are processed to remove motion artifacts with parallelized algorithms, and saved into a memory mapping file format that enables efficient concurrent access. In a second stage, *VolPy* segments candidate neurons using supervised algorithms (Fig 1A and 1C) combined with a manual annotation tool (see S2 Fig and S1 Vid). Finally, *VolPy* denoises fluorescence traces, infers spatial footprints, detects spikes, and extracts subthreshold activity of neurons with parallel processing (Fig 1A and 1D). In Fig 1E and 1F we report the result of running the full pipeline on an example of mouse L1 neocortex voltage imaging dataset. S2 Vid

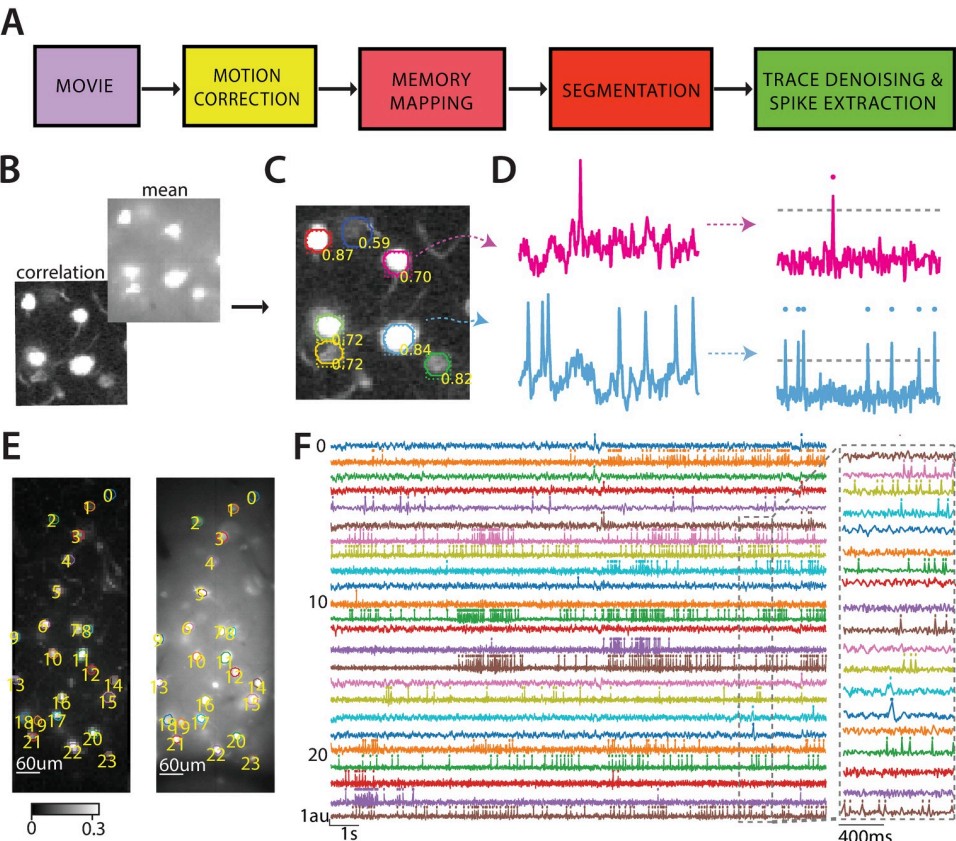

**Fig 1. Analysis pipeline for voltage imaging data.** (A) Four pre-processing steps for segmenting neurons and extracting spikes from voltage imaging movies. (B) The correlation image (back) and the mean image (front) are the inputs to the segmentation step. (C) The segmentation step outputs probabilities of being neurons, bounding boxes and contours. The results are overlaid to the correlation image in (B). (D) Results of trace denoising and spike extraction. Left. The input traces for two neurons in (C). Right. Corresponding denoised traces with better SNR. The dashed horizontal line represents the inferred spike threshold. (E) The correlation image (left) and the mean image (right) of one mouse Layer 1 neocortex dataset with contours detected by *VolPy*. (F) Temporal traces with detected spikes corresponding to neurons in panel (E) extracted by *VolPy* (left). The dashed gray portion of the traces is magnified on the right.

shows the reconstructed movie based on the result of *VolPy* on the same mouse L1 neocortex dataset. S3 Vid provides a demonstration of running the whole pipeline. In what follows we present each stage of the *VolPy* pipeline.

**Motion correction and memory mapping.** First, movies are corrected for sample movement and stored in an efficient format by relying on the infrastructure provided by *CaImAn* [16, 22]. *CaImAn* provides routines to simultaneously register frames to a template and creates

**Table 1. Properties of three heterogeneous types of datasets.** For each type of dataset the name, organism, brain region, source, imaging rate, voltage indicator, and total number of neurons selected by the manual annotators in GT (consensus, (labeler 1, labeler 2, labeler 3)) are given.

| Name | Organism | Brain region | Source | Rate(Hz) | Indicator | # neurons found |
|------|----------|--------------|--------|----------|-----------|-----------------|
| L1 | Mouse | L1 cortex | [5] | 400 | Voltron | 494 (523, 484, 490) |
| TEG | Zebrafish | Tegmental | [5] | 300 | Voltron | 100 (107, 104, 96) |
| HPC | Mouse | Hippocampus | [6] | 1000 | paQuasAr3-s | 66 (72, 86, 73) |

memory-mapped files. Memory mapping provides the ability to quickly read arbitrary portions of the movie in any direction without loading the full movie into memory. In turn, this allows parallelization of all the operations which are required to generate summary images and denoise the signal.

The motion correction algorithm in *CaImAn* is an efficient implementation of NoRMCorre [22]. NoRMCorre is an online algorithm that uses normalized cross-correlation of each frame with a denoised template to infer shifts. Such shifts can be computed either on the overall frame (rigid motion correction) or on patches of the movie (piece-wise rigid motion correction). The latter case is required when movement is non rigid and a simple translation is not sufficient to compensate for the movement. Such algorithm can in many cases be directly applied to the voltage imaging datasets we have considered, because the frames and templates (see for instance Fig 1E) generally contain high-frequency features. Such features are crucial to precisely identify shifts. Importantly, when this fails we allow the option to apply a high-pass spatial filter to help sharpen such features [16]. In *VolPy* the *gSig_filt* parameter controls the size of the kernel for high-pass spatial filtering. We usually inspect visually the results of motion correction, and in all considered cases rigid motion correction was sufficient to capture motion. This might be due to the small size of the field of view (see [22] for a discussion about size of the FOV and its impact on motion correction).

**Segmentation.** SpikePursuit requires as input a set of masks that specify the spatial extent of each neuron, which were provided manually in its initial implementation. With the goal of automating the process and improving consistency across experimenters, we propose to segment neurons with supervised learning approaches. Past attempts at supervised cell localization and segmentation in calcium imaging data have extended U-Net fully convolutional network architectures [23]. In our hands U-Net failed when facing datasets in which ring-shaped neurons overlap (TEG datasets). Other neural network based methods [24–26] have employed 3D representations (width, height and time). However, a popular online calcium imaging segmentation benchmark, Neurofinder (http://neurofinder.codeneuro.org/), reports that these methods are inferior to a well-established neural network architecture for object localization and segmentation, Mask R-CNN [19]. Besides, methods based on 3D representations require spatio-temporal snapshots, whereas Mask R-CNN only requires summary images. Moreover, Mask R-CNN enables separation of overlapping objects in a specific area by providing each object with a unique bounding box.

Mask R-CNN (Fig 2A) provides simultaneous object localization and instance segmentation via a combination of two network portions: backbone and head. The backbone features a pre-trained convolutional network (such as VGG [27], ResNet [28], Inception [29]) for feature extraction. Mask R-CNN also exploits another effective backbone: feature pyramid networks [30], a top-down architecture with lateral connections that enables the network to extract features on multiple scales from the feature maps. In the head, based on the extracted features, a Region Proposal Network proposes initial bounding boxes for each candidate object, which are fed to two downstream branches. One branch is trained to predict a class label with its probability and a bounding box offset which refines the initial bounding box, while the other branch outputs a binary mask for each candidate object. An example of the network inference on a validation dataset by *VolPy* is shown in Fig 1E and 1F.

We adapted Mask R-CNN to our purpose as follows. We chose a combination of ResNet-50 pre-trained on the COCO dataset [31] and feature pyramid networks as the backbone. The input of the network is a three channel image: two for the mean image and one for the correlation image. Three channels are required to match the input to the first few layers pre-trained on the COCO dataset. The network was trained to predict the probability of being a neuron instead of a multi-label output.

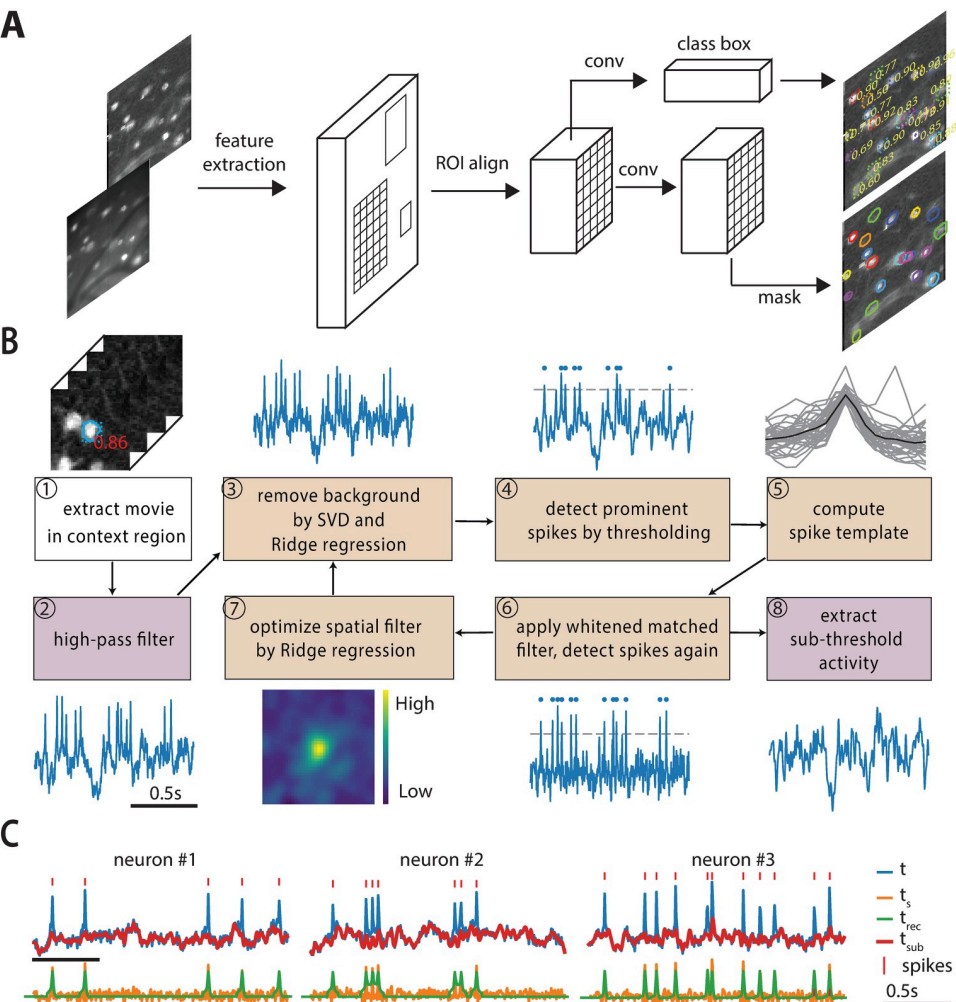

**Fig 2. Segmentation, trace denoising and spike extraction framework.** (A) Mask R-CNN framework for neuron segmentation. The network predicts a probability of being a neuron, a bounding box and a binary mask for each candidate neuron taking summary images as inputs (mean and correlation). (B) Algorithm for fluorescence trace denoising and spike extraction. ①-② Load and high-pass filter the signal in one context region of the movie. The initial temporal trace is computed either by averaging ROI pixels or by applying the spatial filter (if available) to the context region. Two steps are executed in a loop for three iterations. The former (③-⑥) estimates spike times, and the latter (⑦) refines the spatial filter. ③ Extract the first 8 principal components of the background pixels using SVD and then remove the background contamination via Ridge regression. ④ The background-removed trace ($\mathbf{t}$) is high-pass filtered to obtain a zero baseline trace for further processing. Spikes are selected from local maxima higher than the threshold (gray dotted line) using the *adaptive threshold* method. ⑤ Waveforms of these spikes (gray) are averaged to obtain a spike template (black line). ⑥ A whitened matched filter [34] is used to denoise traces and enhance spikes. A second time *adaptive threshold* is applied on the whitened matched filtered trace $\mathbf{t_s}$ to detect spikes. The reconstructed signal ($\mathbf{t_{rec}}$) is obtained by convolving the spike template computed in ⑤ and the inferred spike train. ⑦ Refine the spatial filter through Ridge regression. The product of the context region across time with the refined spatial filter generates the temporal trace for the following iteration. ⑧ After three iterations, the subthreshold activity($\mathbf{t_{sub}}$) is extracted by applying a low-pass filter on the residual trace ($\mathbf{t} - \mathbf{t_{rec}}$). (C) Examples of $\mathbf{t}$, $\mathbf{t_{rec}}$, $\mathbf{t_s}$, and $\mathbf{t_{sub}}$ traces, along with detected spikes.

During training, we randomly cropped the input image into 128x128 patches and applied the following data augmentation techniques using the *imgaug* [32] package: flip, rotation, multiply (adjust brightness), Gaussian noise, shear, scale and translation. Each mini-batch contained six patches. We trained on one GPU the head (the whole network except the ResNet) of

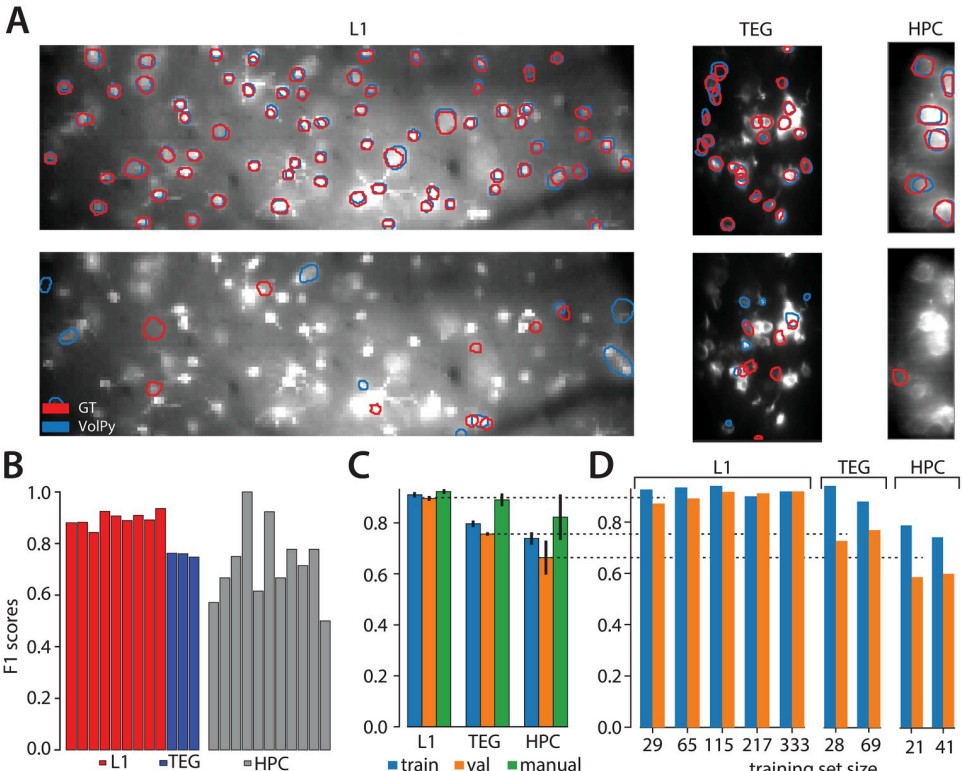

**Fig 3. Evaluation of *VolPy* performance.** (A) Example of *VolPy* segmentation results against three manually annotated datasets (mouse sensory cortex left, larval zebrafish tegmental area center, and mouse hippocampus right). Ground truth was built from three different annotations. Matched and mismatched neurons between *VolPy* (blue) and ground truth (red) were shown in upper and bottom panels respectively. (B) $F_1$ score of *VolPy* for all evaluated datasets in validation. (C) Average $F_1$ score on training and validation sets grouped by dataset type. Results were provided for training, validation and human annotators (against consensus ground truth) (D) Performance of the network in function of training set size for each dataset type.

the network for 20 epochs (2000 iterations) with learning rate 0.01 and then trained the head together with the last 28 layers of the ResNet for another 20 epochs with learning rate 0.001. We used stochastic gradient descent as our optimizer with a constant learning momentum 0.9. The weight decay was 0.0001. It is possible that correlations among RGB channels existing in the original COCO datasets are not present in our datasets, however retraining some of the ResNet layers is likely compensating for this potential issue.

It roughly took 40 minutes to train 40 epochs on a GeForce RTX 2080 Ti GPU with 11 GB of RAM memory. During validation, images were padded with zeroes to make width and height multiples of 64 so that feature maps could be smoothly scaled for the Feature Pyramid Network. We only selected neurons with confidence level greater or equal to 0.7. The output components of the network were further filtered based on the number of pixels in each mask. For TEG datasets, masks containing less than 100 pixels were removed. For HPC datasets, masks containing pixels less than 400 were removed. For L1 datasets, there was no constraint on the number of pixels for each mask. *VolPy* segmentation performance is shown in Fig 3.

While *VolPy* segmentation method achieved good performance on similarly collected datasets, we do not expect it to generalize to completely new datasets out of the box. To overcome this issue, we developed a manual annotation graphical user interface (GUI) tool within *VolPy* to refine the segmentation results. The GUI loads summary images and segmentation results.

It enables users to add/delete neurons and save the results for subsequent steps in the *VolPy* pipeline. We equipped the GUI with one-click semi-automatic neuron segmentation based on the Python Cell Magic Wand Tool [26]. See S1 Vid and S2 Fig for more details. Segmenting neurons on datasets significantly different from the ones we employed to train Mask R-CNN might lead to poor performance. In this case, users may retrain Mask R-CNN based on a step-by-step guide we provide (https://github.com/flatironinstitute/CaImAn/wiki/Training-Mask-R-CNN). Moreover, we also allow users to bypass the Mask R-CNN step and provide their own manual annotations or annotate the data through *VolPy* GUI. This is especially helpful when only few neurons appear in the FOV. We summarize the whole process of segmentation through a flow chart in S3 Fig.

**Trace denoising and spike extraction.**   Classical algorithms for denoising calcium imaging movies and extracting spikes from the corresponding fluorescence traces might underperform when applied to voltage imaging movies. On the one hand, the low signal-to-noise ratio and the complex background fluorescence require new methods for refining spatial footprints, and on the other hand, substantially different biophysical models underlie the temporal dynamics of the fluorescence associated to spikes. To solve both problems, we built upon and extended the SpikePursuit algorithm [5]. We improved SpikePursuit in the following directions: (i) We introduced automatic segmentation of neurons with Mask R-CNN; (ii) We scaled up memory performance thanks to the memory mapping infrastructure; (iii) We scaled up the timing performance by parallel processing and optimizing algorithms; (iv) We introduced a more robust estimate of the background and a simpler option for extracting spikes; (v) We added a method to extract subthreshold signals. In what follows, we introduce SpikePursuit (see Fig 2B), along with our modifications to improve the algorithm. The pseudo-code for the trace denoising and spike extraction routines is shown in Algorithms 1 and 2.

**Loading and preprocessing:**(①② in Fig 2B): As a result of segmentation, each candidate neuron has an associated binary mask which represents its region of interest (ROI). The ROI is dilated to get a larger region centered on the neuron (the context region $C$, ① in Fig 2B). As a first step, all pixels in the context region are efficiently retrieved from the memory mapped file into a 2D matrix $Y \in \mathbb{R}^{T \times N}$, where $T$ is the number of frames and $N = n(C)$ is the number of pixels in the context region $C$. For recordings with indicators featuring signals with reversed polarity (i.e. brighter for lower voltages, such as Voltron), $Y$ is sign-inverted to make spikes positive. $Y$ is high-pass filtered across time with $f_c = 1/3$Hz using a 3rd order Butterworth filter to compensate for photo-bleaching (② in Fig 2B). The high-passed movie is denoted as $Y_h \in \mathbb{R}^{T \times N}$. The initial temporal trace $\mathbf{t} \in \mathbb{R}^T$ of the neuron is computed either as the mean of $Y_h$ over the pixels in the ROI, or – – if a spatial filter $\mathbf{w} \in \mathbb{R}^N$ previously calculated is available – – as the weighted average across all pixels in the context region:

$$\mathbf{t} = \begin{cases} \frac{1}{n(S)} \sum_{x \in S} Y_h(x) & \text{if } \mathbf{w} \text{ is not given} \\ Y_h \mathbf{w} & \text{if } \mathbf{w} \text{ is given,} \end{cases} \tag{1}$$

where $S$ denotes set of pixels in the ROI, $n(S)$ denotes the number of pixels in the ROI, $Y_h(x) \in \mathbb{R}^T$ denotes the signal of $Y_h$ at the pixel x. A spatial filter is a matrix of pixel weights which maximizes the amplitude of extracted spikes calculated as a time-varying weighted sum of pixels in the context region. A good spatial filter may be available from processing another chunk of the movie. Compared to simply averaging pixels across the ROI, the initial trace computed with a spatial filter is expected to have better SNR hence enhance the performance of spike detection.

Afterwards, two steps are iteratively executed for three rounds. The former estimates spike times, and the latter refines the spatial filters.

**Spike time estimation** (③④⑤⑥ **in** Fig 2B): In order to estimate spike times from the fluorescence traces, we first subtract the high-pass filtered movie $Y_h$ from the local background. This step is intended to reduce the chance of the optical cross-talk producing a false spike detection due to an adjacent neuron or process overlapping with the ROI. The local background B is defined as pixels in the context region at least $n_b$ pixels ($n_b$ = 12 by default) away from the ROI. To estimate the influence of the background, we compute the singular value decomposition of the background pixels $Y_b \in \mathbb{R}^{T \times M}$, where $M = n(B)$ denotes the number of pixels in the local background B:

$$Y_b = U\Sigma V, \tag{2}$$

where $U \in \mathbb{R}^{T \times T}$, $V \in \mathbb{R}^{M \times M}$ are two real unitary matrices and $\Sigma \in \mathbb{R}^{T \times M}$ is a rectangular diagonal matrix with non-negative real numbers on the diagonal. We then regress the initial trace **t** on $U_b$ through Ridge regression:

$$\boldsymbol{\beta} = (U_b^T U_b + \lambda_b \|U_b\|_F^2 I)^{-1} U_b^T \mathbf{t}, \tag{3}$$

where $U_b$ is the first $n_{pc}$ (default is 8) components of $U$, $\lambda_b$ (default is 0.01) is the regularization strength and $\beta$ is the estimated ridge regression coefficients. In our experience, many of the largest principal components describe structured global noise in voltage recordings. We choose to subtract the largest 8 components because on real datasets subtracting fewer components would remove less spurious variants, while subtracting many more components would risk subtracting neuronal signals. Compared to SpikePursuit, we add an $L_2$ regularizer to penalize large regression coefficients caused by small components of background pixels with signals bleeding through from the neuron of interest. This provides more reliable results compared to the original linear regression method (see Results section). We choose $\lambda_b$ to be 0.01 because we find that strong regularization strength (greater or equal to 0.1) will not help subtract the background from the signal while small regularization strength will not be able to penalize large regression coefficients enough. We next subtract the background signal from the trace **t** (③ in Fig 2B):

$$\mathbf{t} = \mathbf{t} - U_b\boldsymbol{\beta} \tag{4}$$

The background-removed trace **t** is then filtered with a high-pass fifth order Butterworth filter with $f_c$ = 1Hz, yielding $\mathbf{t_s} \in \mathbb{R}^T$. This operation has the goal of focusing on the frequencies typical of oscillations and spikes. $\mathbf{t_s}$ is then processed for spike extraction (④ in Fig 2B).

Our spike extraction model is based on the idea of matched filters [33, 34]. A matched filter is an optimal linear filter for the detection of a given template from the signal under the hypothesis of additive white Gaussian noise. It has the goal of estimating the location of a given known waveform within a noisy signal while maximizing the SNR. Template matching is performed by correlating a template waveform with the noisy signal. As a consequence, we need two rounds of spike detection. The former is required to prewhiten the signal and form the template waveform. This is used to perform matched filtering by cross-correlating it and the prewhitened trace. The latter consists in identifying and extracting the peaks from the whitened matched filter trace.

To threshold and extract spikes from the filtered signal $\mathbf{t_s}$ we provide two methods, *adaptive* and *simple threshold*. The *adaptive threshold* method selects a threshold ($h$) based on the distribution of local maxima, $P_{max}(x)$, approximated by kernel density estimation. The symmetrization of $P_{max}(x)$ around the median $\mu$ is used to approximate the noise distribution of peaks.

$$\hat{P}_{noise}(\mu + x) = P_{max}(\mu - |x|) \tag{5}$$

The two distributions are then combined to estimate the threshold by minimizing the function:

$$h = \arg\max_{h \in \mathbb{R}} \left( \left( \int_h^\infty P_{max}(x)\, dx \right)^p - \left( \left( \int_h^\infty \widehat{P_{noise}}\, dx \right)^p \right), \right. \tag{6}$$

where $p$ sets the stringency of the discrimination, reflecting a trade off between the benefit of including more (lower-amplitude) spikes in defining the template, and the cost of including additional noise spikes. Smaller values of p result in a more stringent threshold. Although the assumption that the noise distribution of peaks is symmetric around the median $\mu$ is not always fulfilled, experiments with ground truth and simulated data demonstrate that this approach is effective. We set $p = 0.25$ for the first round of spike detection. In our experiments, manual tuning of $p$ was not needed for later stages of the algorithm to identify improved spatial and temporal filters.

In *VolPy*, we provide a second thresholding method, *simple threshold*, which solely relies on the noise level estimation. *simple threshold* only considers values below the median of the filtered trace to estimate the noise level $\sigma$. Only peaks larger than $l$ (default as 3.5) times the noise level $\sigma$ are selected. This method too assumes that the distribution of noise is symmetric around the median. In rigorous terms, the assumption of symmetric distribution is more realistic on the filtered signal than on peak heights. Our rationale to introduce a second thresholding method is to help users with a more intuitive parameter to explore in the cases of *adaptive threshold* failure.

After the first round of spike detection, a spike template $\mathbf{z} \in \mathbb{R}^{2\tau+1}$ is computed by averaging the waveforms of the extracted peaks:

$$\mathbf{z}(t') = \frac{1}{n(\mathbf{s})} \sum_{t \in \mathbf{s}} \mathbf{t_s}(t + t') : t' \in [-\tau, \tau], t' \in \mathbb{Z} \tag{7}$$

where $\mathbf{s}$ is the set of spikes, $n(\mathbf{s})$ is the total number of spikes (⑤ in Fig 2B) and $\tau$ the waveform half size with default time bin of 20 ms (that is 8 frames if the movie was recorded at 400 Hz). Subsequently, a whitened matched filter [34] is used to enhance spikes with shape similar to the template (⑥ in Fig 2B). This operation is composed of two steps: (i) the signal is prewhitened in the frequency domain based on the noise spectrum estimated by the Welch method. The prewhitened signal has noise distribution similar to the white noise which is important as the matched filter is an optimal linear filter when the signal has additive white Gaussian noise. (ii) a new template $\mathbf{z'} \in \mathbb{R}^{2\tau+1}$ is computed from the prewhitened signal (Eq 7) and template matching is performed by computing the cross-correlation between the prewhitened signal and the new template $\mathbf{z}'$. This final signal has peaks associated to spikes enhanced with respect to the original trace.

After the whitened matched filtering operation, a second round of spike detection using *adaptive/simple threshold* is carried out. While in the first round of spike detection $p$ is set to 0.25 in order to avoid False Positives and gather spikes with high confidence to build a representative template, during the second round we aim to maximize $F_1$ score, and therefore set $p = 0.5$. When using the *simple threshold*, a threshold of 3.0 is used by default for a second round of spike detection. The newly detected spikes are transformed into a spike train $\mathbf{q} \in \mathbb{R}^T$. This new spike train is used to reconstruct a denoised version of the original signal, by

convolving $\mathbf{q}$ with the template $\mathbf{z}$:

$$\mathbf{t_{rec}} = \mathbf{q} * \mathbf{z} \quad \text{where } \mathbf{q}(t) = \begin{cases} 1 & \text{if there is a spike at time t} \\ 0 & \text{otherwise} \end{cases} \tag{8}$$

**Spatial filter refinement** (⑦ **in** Fig 2B): The second step is to refine the spatial filter. The updated spatial filter is computed by regressing the reconstructed trace $\mathbf{t_{rec}}$ on the high-passed movie $Y_h$ through Ridge regression:

$$\mathbf{w} = (Y_h^T Y_h + \lambda_w \|Y_h\|_F^2 I)^{-1} Y_h^T \mathbf{t_{rec}}, \tag{9}$$

where $\lambda_w$ (default is 0.01) is the regularization strength. Instead of solving the ridge regression problem in its analytical form as SpikePursuit, we apply an iterative and efficient algorithm [35] implemented in the Scikit-Learn package ('lsqr') for better time performance. Subsequently, the weighted average of the movie with the refined spatial filter is used as the updated temporal trace for the following iteration:

$$\mathbf{t} = Y_h \mathbf{w} \tag{10}$$

For the final round, the spatial filter and the temporal trace are not updated.

**Subthreshold activity extraction** (⑧ **in** Fig 2B): After three iterations of spike time estimation (in our experience this was generally sufficient to converge to a stable solution) and spatial filter refinement, the subthreshold activity is extracted. First, a residual signal is computed by subtracting the reconstructed trace from the temporal trace $\mathbf{t_{res}} = \mathbf{t} - \mathbf{t_{rec}}$. Second, a 5th order Butterworth low-pass filter ($f_c = 20$Hz by default) is applied on the residual trace $\mathbf{t_{res}}$.

**Locality test**: A locality test is performed finally to evaluate whether the reconstructed signal represents the original ROI or is contaminated by neighboring structures. First, we compute the correlation between the reconstructed signal $\mathbf{t_{rec}}$ and each pixel in the movie context region $Y_h$. Second, we check whether the pixel with maximal correlation is inside the original region of interest $S$ or not. In case this did not happen, it would mean that the extracted signal represents other surrounding structures, and therefore it is discarded.

Importantly, inactive neurons are generally identified by the segmentation algorithm, but given the absence of spikes the spatial filters might not match structures internal to the provided masks, thereby failing the locality test. Therefore, inactive neurons with signals not representing the ROI can be removed since they fail locality tests.

```
Algorithm 1 Trace Denoising and Spike Extraction
```

**Require:** Movie in the context region $Y \in \mathbb{R}^{T \times N}$, where T is number of frames and N is number of pixels in the context region, the set of pixels in the region of interest S, the set of pixels in the local background B, the number of selected background principal components $n_{pc}$, the Ridge regression regularization coefficients $\lambda_b$, $\lambda_w$, the number of iterations K, and remaining parameters *params*

1: **if** REVERSEPOLARITYINDICATOR = 1 **then**
2: $Y \leftarrow Y \cdot (-1)$
3: **end if**
4: $Y_h \leftarrow$ HIGHPASSFILTER($Y$, *params*) ▷ Correct for photobleaching
5: **if** w is None **then**
6: $\mathbf{t} \leftarrow \frac{1}{n(S)} \sum_{x \in S} Y_h(:, x)$ ▷ Averaging the signal across pixels in ROI
7: **else**
8: $\mathbf{t} \leftarrow Y_h \mathbf{w}$ ▷ Compute weighted average across all pixels inside the context region
9: **end if**

10: $Y_b \leftarrow Y_h(:, B)$                                    ▷ Extract the background movie
11: $U, \Sigma, V \leftarrow \text{SVD}(Y_b)$                        ▷ Compute the singular value decompostion of $Y_b$
12: $U_b = U(:, 1: n_{pc})$
13: **for** $k \leftarrow 1: K$ **do**
14:    $\beta \leftarrow (U_b^T U_b + \lambda_b \|U_b\|_F^2 I)^{-1} U_b^T \mathbf{t}$            ▷ Ridge regression to remove background
15:    $\mathbf{t} \leftarrow \mathbf{t} - U_b \beta$
16:    $\mathbf{t_s}, \mathbf{s}, \mathbf{t_{rex}}, \mathbf{z} \leftarrow \text{DENOISESPIKES}(\mathbf{t}, params)$            ▷ See Algorithm 2. Compute trace after whitened matched filter $\mathbf{t_s}$, spike time $\mathbf{s}$, reconstructed trace $\mathbf{t_{rec}}$ and spike template $\mathbf{z}$
17:    **if** $k < K$ **then**
18:       $\mathbf{w} \leftarrow (Y_h^T Y_h + \lambda_w \|Y_h\|_F^2 I)^{-1} Y_h^T \mathbf{t_{rec}}$            ▷ Refine spatial filter
19:       $\mathbf{t} \leftarrow Y_h \mathbf{w}$
20:    **end if**
21: **end for**
22: $\mathbf{t_{sub}} \leftarrow \text{LOWPASSFILTER}((\mathbf{t} - \mathbf{t_{rec}}), params)$            ▷ Extract subthreshold activity
23: $m \leftarrow \text{ArgMax}(Y_h^T \mathbf{t_{rec}})$                            ▷ Locality test
24: **if** $m \in S$ **then**
25:   $loc \leftarrow 1$
26: **else**
27:   $loc \leftarrow 0$
28: **end if**
29: **return** $\mathbf{t_s}, \mathbf{t}, \mathbf{t_{sub}}, \mathbf{s}, \mathbf{t_{rec}}, \mathbf{z}, loc$

**Algorithm 2 DenoiseSpikes**
**Require:** Trace $\mathbf{t} \in \mathbb{R}^T$, waveform half size $\tau$, stringency parameter for *adaptive threshold* $p_1$ and $p_2$, threshold parameter for *simple threshold* $l_1$ and $l_2$, and remaining parameters *params*
1: $\mathbf{t_s} \leftarrow \text{HIGHPASSFILTER}(\mathbf{t}, params)$                ▷ Remove low frequency baseline
2: $\mathbf{t_s} \leftarrow \mathbf{t_s} - \text{MEDIAN}(\mathbf{t_s})$
3: **if** $\text{USEADAPTIVETHRESHOLD} = 1$ **then**
4:   $\mathbf{s_1} \leftarrow \text{ADAPTIVETHRESHOLD}(\mathbf{t_s}, p1)$                    ▷ See Algorithm 3
5: **else**
6:   $\mathbf{s_1} \leftarrow \text{SIMPLETHRESHOLD}(\mathbf{t_s}, l_1)$                    ▷ See Algorithm 4
7: **end if**
8: $\mathbf{q_1} \leftarrow \text{ZEROS}(T)$            ▷ Create a zero vector with dimension T
9: $\mathbf{q_1}(\mathbf{s_1}) \leftarrow 1$                            ▷ $\mathbf{q}$ is the spike train
10: $\mathbf{z} \leftarrow \frac{1}{n(\mathbf{s_1})} \sum_{i=1}^{n(\mathbf{s_1})} \mathbf{t_s}(\mathbf{s_1}(i) - \tau : \mathbf{s_1}(i) + \tau)$            ▷ Compute the spike template
11: $\mathbf{t_s} \leftarrow \text{WHITENEDMATCHEDFILTER}(\mathbf{t_s}, \mathbf{q_1}, \mathbf{s_1}, \tau)$                ▷ See Algorithm 5
12: **if** $\text{USEADAPTIVETHRESHOLD} = 1$ **then**
13:    $\mathbf{s_2} \leftarrow \text{ADAPTIVETHRESHOLD}(\mathbf{t_s}, p_2)$
14: **else**
15:    $\mathbf{s_2} \leftarrow \text{SIMPLETHRESHOLD}(\mathbf{t_s}, l_2)$
16: **end if**
17: $\mathbf{q_2} \leftarrow \text{ZEROS}(T)$
18: $\mathbf{q_2}(\mathbf{s_2}) \leftarrow 1$
19: $\mathbf{t_{rec}} \leftarrow \mathbf{q_2} * \mathbf{z}$                ▷ Convolve the spike train with temporal template to get the reconstructed signal
20: **return** $\mathbf{t_s}, \mathbf{s}, \mathbf{t_{rec}}, \mathbf{z}$

**Algorithm 3 AdaptiveThreshold**
**Require:** Trace $\mathbf{t_s} \in \mathbb{R}^T$, stringency parameter $p$, and remaining parameters *params*
1: $\mathbf{p} \leftarrow \text{LOCALMAXIMA}(\mathbf{t_s})$                ▷ Find peak heights of all local maxima

2: $\mathbf{x} \leftarrow$ LinSpace(Min($\mathbf{p}$), Max($\mathbf{p}$), *params*)  ▷ Evenly spaced samples between min and max of peak heights

3: $\mathbf{p_{max}} \leftarrow$ KDE($\mathbf{p}$, $\mathbf{x}$)  ▷ Estimated distribution of local maxima at points $\mathbf{x}$

4: $\mu \leftarrow$ Median($\mathbf{p}$)

5: $j \leftarrow$ Find($\mathbf{x}(i) < \mu$, $\mathbf{x}(i + 1) > \mu$)

6: $\mathbf{p_{noise}} \leftarrow$ Zeros(Len($\mathbf{p_{max}}$))  ▷ Create a zero vector same size as $\mathbf{P_{max}}$

7: $\mathbf{P_{noise}}(1: j) \leftarrow \mathbf{P_{max}}(1: j)$  ▷ Estimate noise distribution by symmetrization

8: **if** $2j \geq$ Len($\mathbf{P_{max}}$) **then**

9:  $\mathbf{P_{noise}}(j + 1:$ end$) \leftarrow \mathbf{P_{noise}}(j: 2j - $ Len($\mathbf{P_{max}}) + 1)$

10: **else**

11:  $\mathbf{P_{noise}}(j + 1: 2j) \leftarrow \mathbf{P_{noise}}(j: 1)$

12: **end if**

13: $\mathbf{F_{max}} \leftarrow$ CumSum($\mathbf{P_{max}}$)  ▷ Cumulative distribution function

14: $\mathbf{F_{noise}} \leftarrow$ CumSum($\mathbf{P_{noise}}$)

15: $\mathbf{F_{max}} \leftarrow \mathbf{F_{max}}($end$) - \mathbf{F_{max}}$

16: $\mathbf{F_{noise}} \leftarrow \mathbf{F_{noise}}($end$) - \mathbf{F_{noise}}$

17: $\mathbf{g} \leftarrow (\mathbf{F_{max}})^P - (\mathbf{F_{noise}})^P$

18: $h \leftarrow \mathbf{x}($ArgMax($\mathbf{g}$))

19: $s \leftarrow$ LocalMaxima($\mathbf{g}$))  ▷ All local maxima with height greater or equal to h

20: **return s**

**Algorithm 4 SimpleThreshold**

**Require:** Temporal trace $\mathbf{t_s} \in \mathbb{R}^T$, threshold parameter l

1: $\mathbf{t}' \leftarrow -\mathbf{t_s}(\mathbf{t_s} < 0)$

2: $\sigma \leftarrow \sqrt{\frac{1}{n(\mathbf{t}')}\sum_{i=1}^{n(\mathbf{t}')}\mathbf{t}'(i)^2}$  ▷ Estimated std based on negative part of the signal $\mathbf{t_s}$

3: $\mathbf{s} \leftarrow$ LocalMaxima($\mathbf{t_s}$, $l \cdot \sigma$)  ▷ Find peaks higher than l times the noise level

4: **return s**

**Algorithm 5 WhitenedMatchedFilter**

**Require:** Temporal trace $\mathbf{t_s} \in \mathbb{R}^T$, spike train $\mathbf{q} \in \mathbb{R}^T$, spike times $\mathbf{s} \in \mathbb{R}^{n(\mathbf{s})}$, waveform half size $\tau$

1: $\mathbf{q}' \leftarrow$ Convolve($\mathbf{q}$, Ones($2\tau + 1$))

2: $\mathbf{t_{noise}} \leftarrow \mathbf{t_s}(\mathbf{q}' < 0.5)$  ▷ The noise signal

3: $\mathbf{s_n} \leftarrow$ Sqrt(Welch($\mathbf{t_{noise}}$))  ▷ $\mathbf{s_n}$ is the scaling factor in the frequency domain

4: $\mathbf{t_s} \leftarrow$ IFFT(FFT($\mathbf{t_s}$)$/\mathbf{s_n}$)  ▷ Scale trace in the frequency domain

5: $\mathbf{z}' \leftarrow \frac{1}{n(\mathbf{s})}\sum_{i=1}^{n(\mathbf{s})}\mathbf{t_s}(\mathbf{s}(i) - \tau : \mathbf{s}(i) + \tau)$  ▷ Compute a spike template based on the prewhitened trace

6: $\mathbf{t_s} \leftarrow$ CrossCorrelation($\mathbf{t_s}$, $\mathbf{z}'$)  ▷ Template matching

7: **return $\mathbf{t_s}$**

## Voltage imaging datasets

**In-vivo datasets.** The datasets we employed were all previously published: Recordings from the tegmental area of larval zebrafish (TEG) and mouse L1 cortex (L1) are described in [5]; Recordings from mouse hippocampus (HPC) are described in [6]. For details about animal protocols and data acquisition refer to the original papers. The name, size and number of labeled neurons for each dataset is reported in Table 2.

**Table 2. All annotated datasets for segmentation of** *VolPy.* For each dataset the name, size of datasets and number of labeled neurons are given.

| Name | Size | # | Name | Size | # |
|------|------|---|------|------|---|
| L1.00.00 | 20000*512*128 | 79 | HPC.29.04 | 20000*164*96 | 3 |
| L1.01.00 | 20000*512*128 | 50 | HPC.29.06 | 20000*228*96 | 2 |
| L1.01.35 | 20000*512*128 | 65 | HPC.32.01 | 20000*256*96 | 7 |
| L1.02.00 | 20000*512*128 | 63 | HPC.38.05 | 20000*176*92 | 4 |
| L1.02.80 | 20000*512*128 | 39 | HPC.38.03 | 20000*128*88 | 5 |
| L1.03.00 | 20000*512*128 | 77 | HPC.39.07 | 20000*264*96 | 6 |
| L1.03.35 | 20000*512*128 | 49 | HPC.39.03 | 20000*276*96 | 7 |
| L1.04.00 | 20000*512*128 | 39 | HPC.39.04 | 20000*336*96 | 5 |
| L1.04.50 | 20000*512*128 | 33 | HPC.48.01 | 20000*224*96 | 8 |
| TEG.01.02 | 10000*364*320 | 31 | HPC.48.05 | 20000*212*96 | 7 |
| TEG.02.01 | 10000*360*256 | 28 | HPC.48.07 | 20000*280*96 | 8 |
| TEG.03.01 | 10000*508*288 | 41 | HPC.48.08 | 20000*284*96 | 4 |

**Voltage imaging with simultaneous electrophysiology datasets.** The simultaneous imaging and electrophysiological data presented in this paper were previously published in [5]. Two of them were extracellular recordings from the TEG area of larval zebrafish, and one intracellular recording from mouse L1 cortex. For details about animal protocol and acquisition refer to the original paper.

**Simulated datasets.** We generated simulated voltage imaging movies modelled upon the L1 dataset examples (Fig 4A). Fluorescence traces were obtained by combining the following components: (i) Spikes times were simulated with an inter spike interval uniformly distributed between 0.1 and 0.2 seconds; (ii) Fluorescence signals associated to spikes were obtained by convolving with a kernel matching the dynamics of Voltron signal in L1 neurons; (iii) Subthreshold activity was simulated by applying a Gaussian filter to white noise; (iv) Fluorescence signals of spikes and subthreshold activity were flipped to match the reverse polarity of the Voltron indicator; (v) To simulate photo-bleaching, the resulting fluorescence signal was modulated with an exponential decaying with a 2500s time constant.

Spatial footprints were simulated as ring shaped and real-valued masks with a small process protruding at different angles (Fig 4A). The shape and size was matched to L1 neurons in real data. The signal associated to each neuron within the movie was obtained by multiplying the simulated fluorescence signal times the spatial footprint. The sum of all neurons represented the imaging movie without background. In order to generate a realistic background signal, we summed the movie without background with a 50x50 pixels patch with no visible neurons from a motion corrected L1 dataset movie. When summing neurons and background, the baseline fluorescence of neurons (brightness) was selected to approximately correspond to *in-vivo* recordings. The spike amplitude was adjusted by changing the amplitude of fluorescence signals with the baseline fluorescence fixed (i.e. changing DF with F fixed). We added out of focus signals for different neurons on our simulated data. The out of focus signal was generated by multiplying the signal of each neuron times a spatial weight computed by applying a Gaussian filter on a randomly selected pixel in the FOV. Finally, white noise was added to every voxel in the simulated data.

For experiments with non-overlapping neurons, we tested neurons with spike amplitudes 0.05, 0.075, 0.1, 0.125, 0.15, 0.175. 0.2. For overlapping cases, we tested neurons with overlapping areas 0%, 6%, 19%, 26% and 35%, and spike amplitudes 0.075, 0.125 and 0.175.

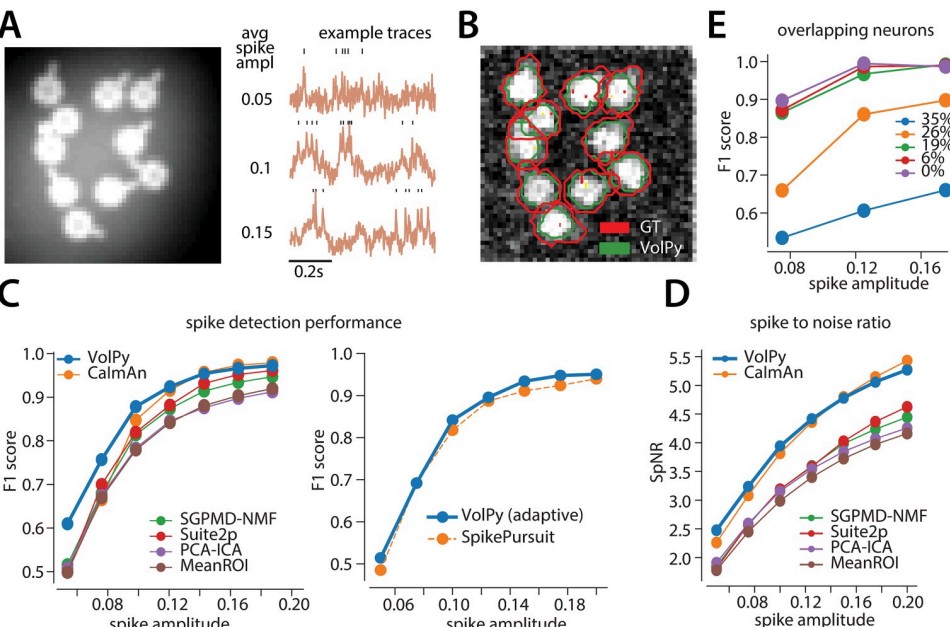

**Fig 4. Evaluation of *VolPy* on simulated data.** (A) Example of simulated data. Left. Average of movie across time. Right. Three example traces with different average spike amplitude. Higher spike amplitude are associated with higher signal to noise ratio. (B) The result of Mask R-CNN in segmenting the simulated movie (0.1 spike amplitude) laying over the correlation image. (C) Performance of *VolPy*, *CaImAn*, SGPMD-NMF, Suite2P, SpikePursuit, PCA-ICA and MeanROI on simulated data. Average $F_1$ score against ground truth in function of spike amplitude. (Left) All algorithms (including *VolPy*) were evaluated with the optimal threshold. (Right) Comparison with SpikePursuit, *adaptive threshold* in both cases. (D) Spike-to-noise ratio (SpNR) in function of spike amplitude. (E) Evaluation of *VolPy* on overlapping neurons. Average $F_1$ score detecting spikes in function of spike amplitude and overlap between two neurons.

## Evaluation of *VolPy*

**Cross-Validation to evaluate the segmentation performance on limited datasets.** In order to decrease the selection bias originated from the separation in training and validation datasets and better evaluate Mask R-CNN model on our limited datasets (24 in total), we performed a stratified three-fold cross-validation. The reason we used a stratified three-fold cross-validation rather than a normal three-fold cross-validation is that we wanted to maintain an equivalent portion of training and evaluation samples for each dataset type. In more detail, we partitioned our datasets into three folds so that L1, TEG, and HPC would all maintain the same number of samples without repetition (Table 3 train/val column shows one group of the partition). During cross-validation two groups were used as training sets while the remaining one as validation set. The cross-validation process was repeated three times with each group used exactly once as validation set.

**Table 3. *VolPy* performance on segmentation.** For each type of datasets, number of datasets, number of neurons, recall, precision, $F_1$ score for training and validation computed by stratified cross-validation are provided.

| Name | #datasets train/val | #neurons train/val | recall(%) train/val | precision(%) train/val | $F_1$(%) train/val |
|------|------|------|------|------|------|
| L1 | 6/3 | 329 ± 32/174 ± 32 | 90 ± 2/88 ± 3 | 92 ± 2/92 ± 3 | 91 ± 1/90 ± 1 |
| TEG | 2/1 | 67 ± 6/33 ± 6 | 78 ± 3/74 ± 4 | 81 ± 1/78 ± 6 | 80 ± 1/76 ± 1 |
| HPC | 8/4 | 44 ± 2/22 ± 2 | 88 ± 2/77 ± 16 | 67 ± 5/61 ± 3 | 74 ± 2/66 ± 7 |

**Precision/Recall framework to measure segmentation performance.**   In order to measure the performance of *VolPy* segmentation, we compared the spatial footprints extracted by *VolPy* with our manual annotations (see [16] component registration for a detailed explanation). In summary, we computed the Jaccard distance (the inverse of intersection over union) to quantify similarity among ROIs, and then solved a linear assignment problem with the Hungarian algorithm [36] to determine matches and mismatches. Once those were identified, we adopted a precision/recall framework and defined True Positive (TP), False Positive (FP), False Negative (FN), and True Negative (TN) as follows:

$$\text{TP} = \text{number of matched spatial footprints}$$

$$\text{FP} = \text{number of spatial footprints in } VolPy \text{ but not in GT}$$

$$\text{FN} = \text{number of spatial footprints in GT but not in } VolPy$$

$$\text{TN} = 0$$

(11)

Next we computed precision, recall and $F_1$ score of the performance in matching as the following:

$$\text{Precision} = \text{TP}/(\text{TP} + \text{FP})$$

$$\text{Recall} = \text{TP}/(\text{TP} + \text{FN})$$

$$F_1 = 2 \times \text{Precision} \times \text{Recall}/(\text{Precision} + \text{Recall})$$

(12)

The $F_1$ score is a number between zero and one. Better performance results in higher $F_1$ score.

For each run of the cross-validation process, we trained a single network and tested it on both training and validation sets. We then computed the mean and standard deviation of the $F_1$ score for different types of datasets with training and validation sets treated separately.

**Precision/Recall framework to measure spike extraction performance with ground truth spikes.**   In order to match spikes extracted from simultaneous voltage imaging and electrophysiology datasets, we employed a greedy matching algorithm. Let **v** and **e** be two sequences of spike times extracted from voltage imaging and electrophysiology datasets respectively. We started by matching the leftmost spike of **v** and **e**. Without loss of generality, we assumed the leftmost spike is **v**(1). If the distance between spike **v**(1) and its closest spike **e**(1) in the other sequence was within 10ms, then two spikes were matched and removed from the sequences; otherwise, spike **v**(1) was considered a mismatch and removed from the sequence **v**. We then started to match the following leftmost spike. This process was repeated until there is no spike in any of these two sequences left. We chose to match spikes within 10 ms based on the fact that neighboring spikes in the electrophysiology had a minimum inter-spike interval of 30 ms.

After identifying matches and mismatches, we proceeded similarly to what explained above. We defined TP, FP, FN, TN similar to Eq 11:

$$\text{TP} = \text{number of matched spikes}$$

$$\text{FP} = \text{number of spikes in } VolPy \text{ but not in GT}$$

$$\text{FN} = \text{number of spikes in GT but not in } VolPy$$

$$\text{TN} = 0$$

(13)

Then we calculated the $F_1$ score same as Eq 12.

**Spike-to-noise-ratio.**   In order to compare the performance of different algorithms when no ground truth data is available, we have defined a metric to quantify how well an algorithm was able to increase the detectability of spikes in voltage imaging traces. When comparing two algorithms, the *Spike-to-noise-ratio* (SpNR) is computed by identifying the set of spikes which were detected by both algorithms, and then calculate, for each denoised trace independently, the ratio between the average spike amplitude and the noise estimated as the standard deviation of the negative portions of the 15Hz high-pass filtered signal. We believe this metric is independent on thresholding methods and should provide an unbiased estimate of an algorithm denoising capability.

## Implementation of benchmarked algorithms

We compared *VolPy* against a set of other algorithms. Some of them could not directly be applied to voltage imaging, and therefore we had to introduce some modifications to adapt them. In what follows we describe how we deployed each of them.

**CaImAn.**   *CaImAn* is a software package for the analysis of calcium imaging data [16] and can be found at the github repository https://github.com/flatironinstitute/CaImAn. For voltage imaging movie using indicator with reversed polarity (i.e. brighter for lower voltages, such as Voltron), vanilla *CaImAn* failed to retrieve reasonable spatial or temporal components because the NMF framework was unable to extract negative spikes of voltage signals. Just flipping the signal and removing the minimum of the whole movie also leads to poor performance. The best results were obtained by flipping the signal and removing the minimum value of each pixel. This helps *CaImAn* focus on the variance related to the voltage signal and not on baseline fluctuations. This modified movie can be processed via the standard *CaImAn* pipeline. We used the *greedy roi* method for spatial footprint initialization. We turned off the deconvolution step used for calcium signals, and instead we high-passed the temporal components with a 15Hz filter and applied a manual threshold to extract spikes.

For simulations, as movies were simulated with negative spikes, we processed them in the same way as we did for movies using Voltron indicator.

For voltage imaging movies using the paQuasar indicator, as they had positive spikes, we passed the original movie directly into the *CaImAn* pipeline without preprocessing and performed the same spike extraction step as mentioned before.

**MeanROI.**   Region of interests were provided from ground truth masks beforehand. The MeanROI method extracts the voltage signal for each neuron by averaging the pixels within the provided masks. Depending on the signal polarity the trace can be flipped. Analogously to *CaImAn*, spikes are extracted by high-passing the signals with a 15Hz filter and manual thresholding.

**Suite2P.**   Suite2p is a software package for the analysis of calcium imaging data [17]. We tested Suite2P on simulated datasets using the software available on github https://github.com/MouseLand/suite2p. To obtain good spatial footprints, we set the *overlapping* parameter to be True across simulations. When the spike amplitude was lower or equal to 0.01, we turned off the *sparse_mode* and when the spike amplitude was 0.005, we turned off the *connected* parameter. We flipped and high-passed the signals with a 15Hz filter and applied a manual threshold to extract spikes.

**SGPMD-NMF.**   SGPMD-NMF is a set of software packages that can be deployed for the analysis of voltage imaging datasets [11, 12]. We obtained SGPMD-NMF from the Github repositories https://github.com/adamcohenlab/invivo-imaging and https://github.com/m-xie/trefide.git. We fed the original movie into the denoising step [11]. We flipped the output signal if it had reverse polarity and passed it to the demixing step, which output spatial footprints and

temporal traces, along with subthreshold activities [12]. We high-passed the signals with a 15Hz filter and applied a manual threshold to extract spikes.

**PCA-ICA.**   We implemented the PCA-ICA algorithm based on previous work on calcium imaging [14]. We chose the top 50 principal components in PCA and 15 components in spatial-temporal ICA. The parameter which controls the relative contribution of spatial and temporal information was set to 0.05. The output spatial components were updated after Gaussian smoothing and thresholding. Temporal signals were extracted as the weighted average of the updated spatial components. We flipped and high-passed the signals with a 15Hz filter and applied a manual threshold to extract spikes.

**SpikePursuit.**   We recovered the original SpikePursuit implementation in Matlab from the github repository https://github.com/KasparP/SpikePursuitMatlab and ran it on simulated data and scalability tests. Ground truth masks were provided as the input and the algorithm used the *adaptive threshold* method to automatically select the optimal threshold and spikes times. For the scalability tests, parameters were set similarly to *VolPy*.

**Spike extraction.**   In this paper we tested the algorithms above and compared their performances against *VolPy*. Since only *VolPy* and SpikePursuit were able to extract spikes automatically, we modified the other algorithms to automate the spike detection process, and thus provide a direct comparison with *VolPy*. The general process to extract spikes was to high-pass the temporal components extracted by each algorithm with a 15Hz filter, and then to apply a manual threshold to extract spikes. The manual threshold was an estimated level of standard deviation of the signal based on the negative portion of the signal (similarly to SNR measure for calcium traces in [16]). In simulations, instead of picking a single manual threshold (for example 3.0), for each spike amplitude value we performed a grid search (range 2.0-4.0 with interval 0.1), and selected the threshold outputting the best average $F_1$ score across all neurons. In simultaneous electrophysiology and voltage imaging datasets and in-vivo datasets, the thresholds for *CaImAn*, MeanROI and SGPMD-NMF were chosen manually. In in-vivo datasets, the SpNR was computed only on the intersection of detected spikes among different algorithms.

**Spatial footprint matching.**   Using the same way to match the segmentation results with manual annotations as mentioned above, spatial footprints of benchmarked algorithms (except SpikePursuit and MeanROI which needed masks beforehand) were matched with manual annotations in simulations. $F_1$ score and SpNR were computed only on the intersection of matched neurons among different algorithms.

## Results

In what follows we report a systematic evaluation of *VolPy* against ground truth segmentation and against other algorithms in terms of performance in spike extraction and scalability.

### *VolPy* performance in localizing neurons

We trained the Mask R-CNN neural networks in *VolPy* on all three types of datasets at the same time (Table 1). We evaluated the networks with a 3-fold cross-validation (see Materials and methods for details). In Fig 3A, we compared the contours predicted by *VolPy* (blue) with manual annotations (red) on three example datasets: *VolPy* was able to identify candidate neurons in conditions of low signal-to-noise and spatial overlap. In Fig 3B and 3C, we quantified *VolPy*'s performance in segmentation using a precision/recall framework and in Table 3 we summarized the average $F_1$ score separately for those three types of datasets. For validation, *VolPy* obtained $F_1$ score of 0.90 ± 0.01 on the L1 datasets (trained with 329 neurons on average), 0.76 ± 0.01 on the TEG datasets (trained with 67 neurons on average), and 0.66 ± 0.07 on

the HPC datasets (trained with 44 neurons on average). To benchmark *VolPy*'s segmentation performance, in Fig 3C, we also reported the average $F_1$ score of all labelers (green bars) when comparing their original annotations against the consensus ground truth: 0.92 ± 0.01 on the L1 datasets, 0.89 ± 0.02 on the TEG datasets and 0.82 ± 0.09 on the HPC datasets.

There were substantial differences in the performance among the three types of datasets for *VolPy*, with L1 obtaining excellent results (close to human annotators) and TEG and HPC progressively worse results (Fig 3C). We hypothesized that two possible sources of variability could account for these differences: the objective difficulty in segmenting the datasets and the number of neurons for training. With regard to segmenting difficulty, the differences among these three types of datasets could be seen clearly through the performance of manual annotators, in which L1 yielded the highest $F_1$ score and HPC yielded the lowest one, and with the largest variance. With regard to the number of neurons for training, in average 329 neurons in L1 datasets were used for training compared to only 67 and 44 neurons in TEG and HPC datasets respectively (see Table 3). To test whether differences in the number of neurons for training mainly account for the variability, we separately trained a network for each type of dataset and varied the training set size (i.e. number of neurons for training, see Fig 3D). Although most likely overfitting was present with less than ∼100 neurons (See S4 Fig), *VolPy* still achieved 0.88 $F_1$ score on the validation sets of L1 when trained with only 29 neurons. We observed that increasing training set size moderately helped improve *VolPy*'s performance on all three types of datasets. Our results suggest that the objective difficulty accounts for most of the difference in performance within the tested range. However, considering that very small training set size of TEG and HPC were used for training, it is likely that *VolPy*'s performance on these two types of datasets may further increase when trained with more neurons.

### *VolPy* performance in spike detection

We validated the spike extraction performance of *VolPy* and other algorithms on the following simulated datasets and real datasets: 1. simulations based on mouse cortex datasets (L1); 2. simultaneous voltage imaging and electrophysiology datasets from mouse neocortex (L1) and zebrafish tegmental area (TEG); 3. Real datasets from mouse neocortex(L1), zebrafish tegmental area (TEG) and mouse hippocampus (HPC). We quantified the performance of the benchmarked algorithms based on precision/recall and *spike-to-noise-ratio* (SpNR, a metric quantifying spike detectability, see Materials and methods for details). Below we detail the comparisons we have performed.

**VolPy performance on simulated data.**    As explained in the Materials and methods section, we simulated voltage imaging movies based on the L1 datasets. The amplitude of spikes in such simulated neurons were varied to model different signal-to-noise ratio cases (Fig 4A). We compared *VolPy* against other algorithms including *CaImAn* [16], SGPMD-NMF [12], Suite2P [17], PCA-ICA [14], MeanROI and SpikePursuit [5] (See Materials and methods for a detailed implementation of each algorithm).

An example of *VolPy* segmentation result is reported in Fig 4B. We did not compare the performance of *VolPy* in segmenting neurons with other considered algorithms for two reasons. First, *CaImAn*, PCA-ICA, Suite2P and SGPMD-NMF only segment active neurons and need manual post processing. Indeed, depending on the input parameters, these algorithms may generate spurious components that cause false positives, making a fair comparison difficult. Second, segmentation on our simulated data is a relatively easy task: all algorithms managed to find all 10 neurons (recall) when spike amplitude was greater or equal to 0.075.

To benchmark *VolPy*'s spike extraction performance against other algorithms that do not have automatic spike thresholding, we selected an optimal threshold by grid search after

customized signal filtering for each of these algorithms (including *VolPy*, see Materials and methods). The idea in this case is to assess how algorithms work independently of thresholding parameters. We therefore assessed the $F_1$ score and SpNR of different algorithms under various SNR scenarios on a FOV containing 10 non-overlapping neurons (Fig 4C and 4D). The results showed that *VolPy* achieved a better $F_1$ score and SpNR in most cases, especially in low-SNR settings. This was expected because *VolPy* optimized spatial footprints and enhanced spike amplitudes through the whitened matched filter. As a control, *VolPy* showed a significantly better result than MeanROI, that is simply averaging the pixels within the ROI. Spike-Pursuit adopts an adaptive threshold method which does not need manual thresholding and was therefore directly compared to VolPy with adaptive threshold. *VolPy* performed slightly better than SpikePursuit (Fig 4C), mainly due to the modifications of using ridge instead of linear regression for background subtraction.

On separate simulations, we evaluated the performance of *VolPy* in the case of overlapping neurons. We simulated movies with two overlapping neurons and assessed the $F_1$ score for *VolPy* when varying the degree of overlap (Fig 4E). In terms of spatial footprint extraction, *VolPy* started failing to segment neurons in the large overlapping case (35%). The result could potentially be improved if Mask R-CNN was trained on the simulated datasets as well, especially on neurons featuring similar overlap. Given this, we provided manual masks for testing *VolPy* in such simulated overlapping scenarios. In terms of spike detection, *VolPy* maintained good results when the total overlap was less than 20%. However, the $F_1$ score dropped as the overlapping area further increased.

**_VolPy_ performance on simultaneous voltage imaging and electrophysiology data.**   We tested *VolPy*, *CaImAn*, MeanROI and SGPMD-NMF on datasets with simultaneous voltage imaging and electrophysiology. We used three recordings from mouse L1 neocortex and Zebrafish Tegmental area [5]. Spikes for electrophysiology recordings were obtained by manual thresholding. We automatically analyzed voltage imaging data with the different algorithms. For the *VolPy* pipeline, we used the *adaptive threshold* method and the outputs were the spatial footprints, the voltage traces, and the corresponding spike times (Fig 5A). For the other algorithms, we used manual thresholding for spike extraction. *VolPy* was able to extract good spatial footprints in all cases while *CaImAn* and SGPMD-NMF failed to output good spatial footprints in two (fish datasets) of these three datasets (See S5 Fig). As a consequence, *VolPy* achieved the best results in terms of $F_1$ score (Fig 5B), whereas *CaImAn* and SGPMD-NMF produced worse results than MeanROI in all datasets. We believe that *CaImAn* and SGPMD-NMF underperformed because neurons in voltage imaging movies were not firing with homogeneous spatial footprints, as could be observed from S4 Vid. This problem was exacerbated in the high spatial resolution recordings, where neurons are represented by a larger number of pixels. Moreover, we observed that the denoising step in SGPMD sometimes reduced the SNR on noisy datasets.

**Comparison of _VolPy_ and other methods on real datasets.**   Finally, we compared the performance of *VolPy* with other algorithms on the L1 (8 neurons), HPC (2 neurons) and TEG (1 neuron) datasets. Since the ground truth was missing in these cases we quantified how each algorithm was able to enhance spike detection. We computed the SpNR metric only on spikes detected by all algorithms. In Fig 5C we presented some example neurons and the corresponding traces extracted by *VolPy*, *CaImAn*, MeanROI and SGPMD-NMF. In Fig 5D, SpNR was computed for each algorithm across different datasets.

Taken together, the results above confirm that *VolPy* outperforms existing algorithms in terms of spike extraction.

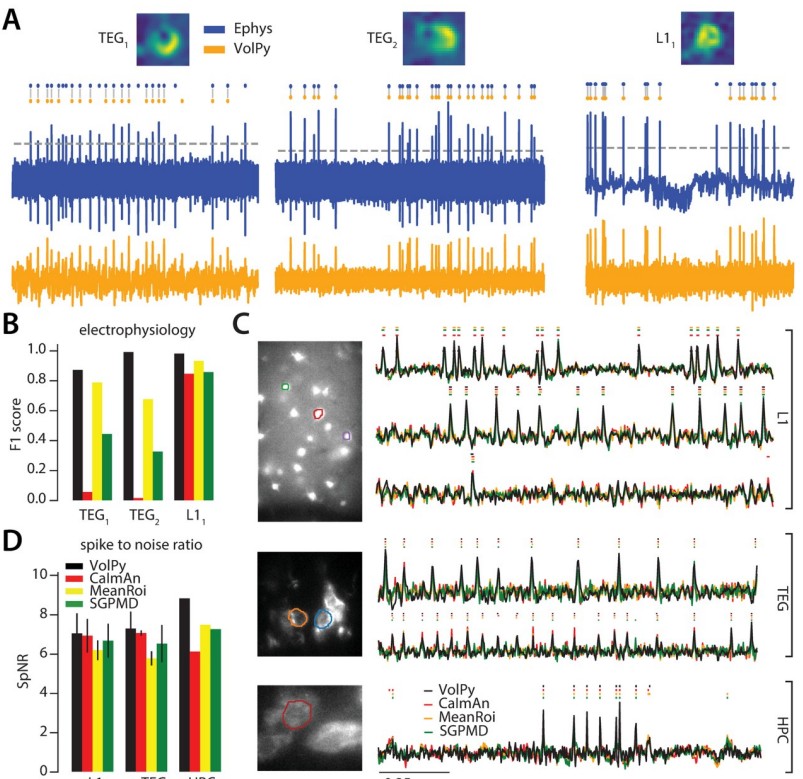

**Fig 5. *VolPy* performance on real data.** (A) Evaluation of *VolPy* spike extraction performance against simultaneous electrophysiology. Three neurons, two from larval zebrafish TEG area ($TEG_1$ and $TEG_2$) and one from mouse L1 ($L1_1$), for which we had available both electrophysiology and imaging. Top. Spatial footprint extracted by *VolPy*. Middle. Ground truth spikes from electrophysiology (blue) and spikes extracted by *VolPy* (orange), gray vertical lines indicate matched spikes. Bottom. Electrophysiology (blue, top) and fluorescence signal denoised by *VolPy* (bottom, orange). (B) We compared the performance of *VolPy*, *CaImAn*, MeanROI, and SGPMD-NMF in retrieving spikes on the three neurons in (A). (C) Examples of trace extraction results for *VolPy*, *CaImAn*, MeanROI, and SGPMD-NMF. On the left mean image overlaid to example neurons (top L1, middle TEG, bottom HPC). On the right traces and inferred spikes for datasets L1 (top three traces), TEG (traces 4-5 from top) and HPC (bottom trace). (D) Spike to noise ratio (SpNR) for each considered algorithm and dataset type.

## *VolPy* scalability

We examined the performance of *VolPy*, SGPMD-NMF and SpikePursuit in terms of processing time and peak memory for the L1 datasets presented above. We ran our tests on a linux-based desktop (Ubuntu 18.04) with 16 CPUs (Intel Core i9-9900K CPU 3.60GHz) and 64 GB of RAM. For segmentation, we used a GeForce RTX 2080 Ti GPU with 11 GB of RAM memory. An L1 movie (FOV 512 × 128, pixels × pixels) with 75 annotated neurons was used for all our scalability test.

Fig 6A reports *VolPy* processing time in function of the number of frames using 8 processors. The results showed that the processing time scales linearly in the number of frames. Processing 75 candidate neurons in the 1.6 minutes long movie (40000 frames) took about 8 minutes. Spike extraction (red bar) was the most time consuming step. In order to probe the benefits of parallelization, we ran *VolPy* 4 times while limiting the available CPUs to 1, 2, 4 and 8 on 40000 frames of the movie (Fig 6B). We observed significant performance gains due to parallelization, especially in the motion correction and spike extraction phase, with a maximum speed-up of 3X.

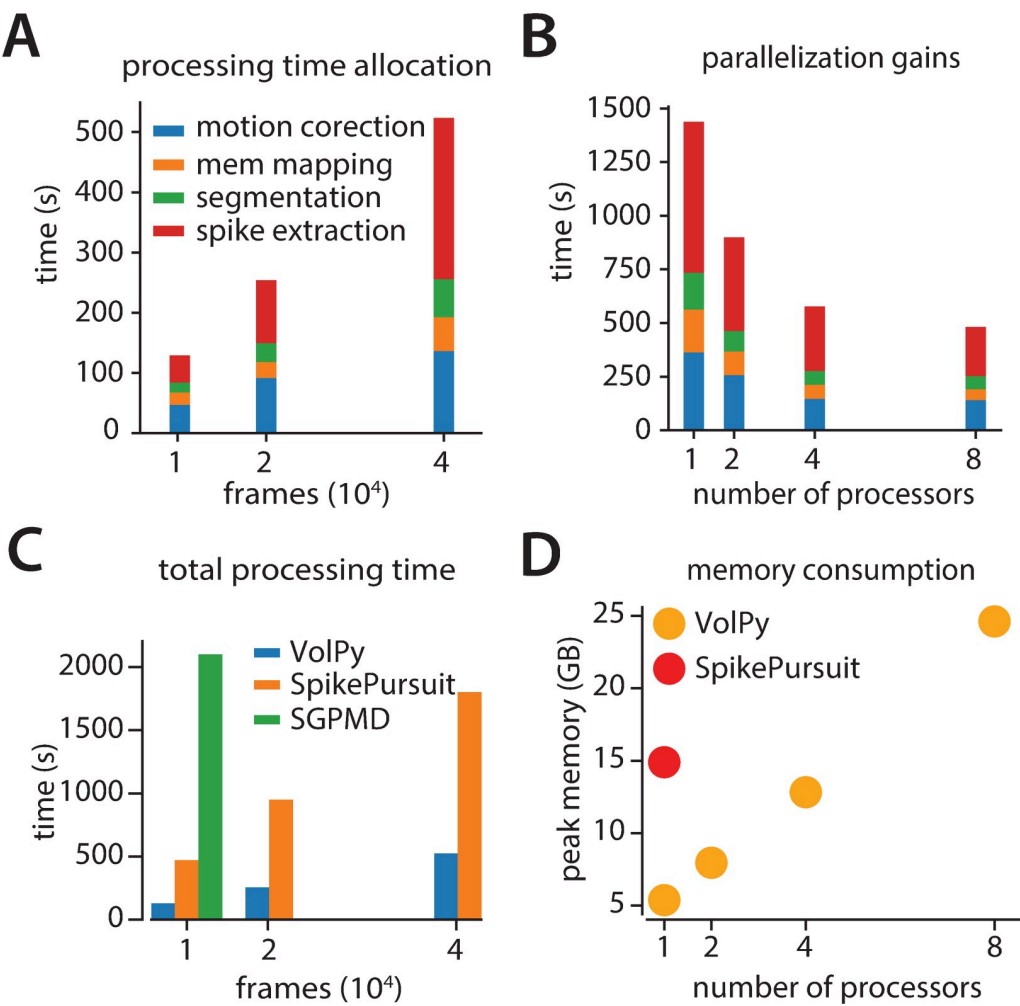

**Fig 6. Evaluation of *VolPy* scalability.** *VolPy* scalability was evaluated based on a 512x128 pixels movie with 75 annotated neurons. (A) Processing time allocation of *VolPy* with 10000, 20000 and 40000 frames using 8 processors. (B) Processing time of *VolPy* on 40000 frames with 1, 2, 4 and 8 processors. (C) Comparison of performance among *VolPy* (8 processors), SpikePursuit and SGPMD-NMF with 10000, 20000 and 40000 frames. (D) Peak memory usage of *VolPy* and SpikePursuit on 40000 frames. Since *VolPy* supports parallelization we reported memory usage with 1, 2, 4 and 8 processors.

In order to assess whether *VolPy* improves over other algorithms in terms of memory and speed, we benchmarked its performance against SpikePursuit and SGPMD-NMF in function of the number of frames(Fig 6): SpikePursuit was about 3.5X slower and SGPMD-NMF was about an order of magnitude less performing. The speed gains of *VolPy* against SpikePursuit is mainly due to parallel processing and optimizing the SpikePursuit algorithm. In terms of memory usage (Fig 6D), SGPMD-NMF produced an out of memory error when processing 20000 frames on our machine, whereas SpikePursuit consumed about 3 times the amount of memory compared to *VolPy* (both using a single CPU on a 40000 frames movie).

In conclusion, *VolPy* significantly improves scalability over existing approaches for voltage imaging data analysis.

## Discussion

Here we introduced *VolPy*, the first end-to-end, automatic, scalable, and open source analysis pipeline for large scale voltage imaging datasets. *VolPy* corrects movies for motion artifacts, automatically detects neurons, and it provides trace denoising and adaptive spike time estimation based on robust and scalable peak detection algorithms. Finally, *VolPy* is embedded into the popular open-source software *CaImAn*, and therefore directly available to the existing community.

### *VolPy* evaluation

We evaluated *VolPy* segmentation performance against a corpus of manually annotated datasets obtained from three human labelers. Humans generally agreed well with a consensus ground truth, with more consistent labeling for easy and high SNR datasets. More difficult datasets, such as HPC, produced controversial annotations and less agreement. On the tested datasets, Mask R-CNN quickly reached asymptotic performance even using small training sets (∼30 neurons), and that performance did not seem to dramatically improve with larger training sets. Our tests suggest that the objective difficulty is the main responsible for performance degradation, with training set size modestly affecting the performance in the tested range. We cannot exclude that a very large corpus of annotated datasets might increase the segmentation performance on difficult datasets, such as HPC. As more dataset become available we plan to further test this possibility.

We benchmarked the accuracy of *VolPy* in detecting spikes against other algorithms and demonstrated that *VolPy* in most cases outperforms these methods, especially in low SNR scenario, typical for voltage imaging. Finally, *VolPy* is faster and consumes less memory than all the tested voltage imaging analysis algorithms. Taken together, our experiments indicate that *VolPy* is a competitive package for the analysis of voltage imaging data, and we expect it to become a useful tool to researchers, as the advances in voltage indicators spread within the neuroscience community.

### Practical improvements

Since the segmentation results from *VolPy* might be imperfect on some dataset types, we provide a graphical interface within *VolPy* to refine the output of Mask R-CNN manually (S1 Vid) or annotate new datasets. In conjunction, we provide a step-by-step guide to train Mask R-CNN neural networks with newly labelled datasets (https://github.com/flatironinstitute/CaImAn/wiki/Training-Mask-R-CNN).

### Future extensions

As more data will become available and more users will adopt *VolPy*, we plan to develop a web-based graphical user interface for experimentalists to manually segment datasets and transfer the resulting annotations to a distributed computing server, which will periodically retrain the network and improve the performance of our system. Currently, only a small number of labs have publicly shared cellular-resolution large-scale voltage recordings. However, we anticipate that these methods will rapidly become more widely adopted, resulting in a much wider variety of dataset types.

In the future, we plan to extend this framework in two algorithmic directions. First, similar to calcium imaging [37], we plan to develop methods appropriate to real-time scenarios, where activity of neurons needs to be inferred on-the-fly and frame-by-frame; Second, we plan

to include algorithms appropriate to quantify supra- and subthreshold signals in neuronal sub-compartments.

## Availability

The code and datasets are available at the master branch of the repository https://github.com/flatironinstitute/CaImAn and at the Zenodo repository [38]. Data used in figures are included in S1 Data. Mask R-CNN is available (only required for retraining the network), from https://github.com/matterport/Mask_RCNN. Mask R-CNN was trained with the following tools: python 3.7.3, tensorflow-gpu 1.14.0.

## Supporting information

**S1 Data. Data for generating figures.** An excel file contains data for generating Figs 3B, 3C, 3D, 4C, 4D, 4E, 5B, 5D, 6A, 6B, 6C and 6D.
(XLSX)

**S1 Fig. Manual annotations of voltage imaging datasets with ImageJ.** We selected neurons based on mean image (left), correlation image (mid). Three annotators marked the contours of neurons independently using ImageJ Cell Magic Wand tool plugin and showed selections in ROI manager (right).
(TIF)

**S2 Fig. Manual annotation *VolPy* GUI.** The interface helps user to select neurons either using polygons (point by point) or a Python implementation of the ImageJ Cell Magic Wand [26]. Users can then remove or add masks, and finally save in hdf5 format the output.
(TIF)

**S3 Fig. Flow chart for segmentation.** Summary images are computed from input voltage imaging movies. Subsequently masks of neurons can be provided in two ways. 1. Neurons can be segmented via a Mask R-CNN neural network trained on the three types of datasets presented in this paper (L1, TEG and HPC). The output labels can be further corrected by the *VolPy* GUI (See S1 Vid). If users are not satisfied with results of Mask R-CNN, they can manually annotate voltage imaging datasets using ImageJ. Such new annotations can then be used to retrain Mask R-CNN. Details for retraining Mask R-CNN are explained at the page https://github.com/flatironinstitute/CaImAn/wiki/Training-Mask-R-CNN 2. Users can also bypasss the Mask R-CNN step and choose to provide their own manual masks labelled either through other softwares or *VolPy* GUI.
(TIF)

**S4 Fig. Learning curves in function of dataset size.** Learning curves corresponding to data in Fig 3D. Training (blue) and validation (orange) loss in function of training set size for L1 (A), TEG (B) and HPC (C) datasets. (D) For comparison, learning curves for training and validation set when training on all the datasets.
(TIF)

**S5 Fig. Spatial footprint for simultaneous imaging and electrophysiological data.** Spatial footprints extracted by *CaImAn* and SGPMD-NMF on the data reported in Fig 5A.
(TIF)

**S1 Vid. Example of manual annotation interface in *VolPy*.** This is used to annotate datasets quickly when there are few neurons in FOV.
(MP4)

**S2 Vid. Example of reconstructed movie in *VolPy*.** Left: movie after motion correction. Middle: movie with baseline removed. Right: reconstructed movie.
(MP4)

**S3 Vid. The whole *VolPy* pipeline.** Demonstration of what it looks like running the complete pipeline.
(MP4)

**S4 Vid. Neurons feature non-stationary spatial components.** The movie provides an example of how a neuron might not be easily represented by a single spatial component, since it is not always the same portion of the neuron that becomes brighter during voltage changes.
(MOV)

## Acknowledgments

We thank K Svoboda, M Ahrens, T Kawashima, Y Shuai, A Cohen, M Xie for providing voltage imaging datasets. We thank Jimmy Tabet and Maddison Khire for annotating the datasets.

## Author Contributions

**Conceptualization:** Changjia Cai, M. Hossein Eybposh, Eftychios A. Pnevmatikakis, Kaspar Podgorski, Andrea Giovannucci.

**Data curation:** Changjia Cai, Johannes Friedrich, Amrita Singh, Kaspar Podgorski, Andrea Giovannucci.

**Formal analysis:** Changjia Cai, Kaspar Podgorski, Andrea Giovannucci.

**Funding acquisition:** Kaspar Podgorski.

**Investigation:** Changjia Cai, Kaspar Podgorski, Andrea Giovannucci.

**Methodology:** Changjia Cai, Johannes Friedrich, Amrita Singh, Eftychios A. Pnevmatikakis, Kaspar Podgorski, Andrea Giovannucci.

**Project administration:** Andrea Giovannucci.

**Resources:** Amrita Singh.

**Software:** Changjia Cai, Johannes Friedrich, Amrita Singh, M. Hossein Eybposh, Eftychios A. Pnevmatikakis, Andrea Giovannucci.

**Supervision:** Kaspar Podgorski, Andrea Giovannucci.

**Validation:** Changjia Cai, Johannes Friedrich, Kaspar Podgorski, Andrea Giovannucci.

**Visualization:** Changjia Cai.

**Writing – original draft:** Changjia Cai, Andrea Giovannucci.

**Writing – review & editing:** Changjia Cai, Johannes Friedrich, Amrita Singh, M. Hossein Eybposh, Eftychios A. Pnevmatikakis, Kaspar Podgorski, Andrea Giovannucci.

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
