## [Decision Letter · Decision Letter 0]

30 Jun 2020

Dear Dr. Giovannucci,

Thank you very much for submitting your manuscript "VolPy: automated and scalable analysis pipelines for voltage imaging datasets" for consideration at PLOS Computational Biology.

As with all papers reviewed by the journal, your manuscript was reviewed by members of the editorial board and by several independent reviewers. In light of the reviews (below this email), we would like to invite the resubmission of a significantly-revised version that takes into account the reviewers' comments.

Both reviewers agree on the need for analysis tools for voltage-imaging data. Due to the nature of the underlying signals and the reporters, these data do pose a significant challenge for analysis, to detect real signals, disambiguate the cellular sources etc. This is obviously an evolving field since breakthroughs in reporters will change the requirements for analysis, hopefully making life easier. Both reviewers agree that the approaches taken in this article are reasonable given the data sets that they consider. The proposed pipeline is a collection of previously described approaches and techniques, not new techniques.

Both reviewers agree that the paper would have far higher impact if it included comparisons to existing methods, some of which have been tested on voltage imaging data. I agree and comparisons should be made to the tools mentioned by reviewer 1. Claims of validity of methods were based on a small number of annotators, so again, it is crucial to check claims and expected behavior against other methods and revise either the methods or claims if and when unexpected results appear.

There are also a large number of specific questions and concerns from reviewer 1, which must be addressed point by point.

Please also answer the following questions from the editor: 1. Were surrogate data sets used or can they be used to test validity of the methods? 2. Were model data sets with known ground truth spike times, cell identities, etc. constructed or used or can they be used to test the validity of methods?

We cannot make any decision about publication until we have seen the revised manuscript and your response to the reviewers' comments. Your revised manuscript is also likely to be sent to reviewers for further evaluation.

Sincerely,

Corey Acker

Guest Editor

PLOS Computational Biology

Kim Blackwell

Deputy Editor

PLOS Computational Biology

Both reviewers agree on the need for analysis tools for voltage-imaging data. Due to the nature of the underlying signals and the reporters, these data do pose a significant challenge for analysis, to detect real signals, disambiguate the cellular sources etc. This is obviously an evolving field since breakthroughs in reporters will change the requirements for analysis, hopefully making life easier. Both reviewers agree that the approaches taken in this article are reasonable given the data sets that they consider. The proposed pipeline is a collection of previously described approaches and techniques, not new techniques.

Both reviewers agree that the paper would have far higher impact if it included comparisons to existing methods, some of which have been tested on voltage imaging data. I agree and comparisons should be made to the tools mentioned by reviewer 1. Claims of validity of methods were based on a small number of annotators, so again, it is crucial to check claims and expected behavior against other methods and revise either the methods or claims if and when unexpected results appear.

There are also a large number of specific questions and concerns from reviewer 1, which must be addressed point by point.

Please also answer the following questions from the editor: 1. Were surrogate data sets used or can they be used to test validity of the methods? 2. Were model data sets with known ground truth spike times, cell identities, etc. constructed or used or can they be used to test the validity of methods?

Reviewer's Responses to Questions

**Comments to the Authors:**

Reviewer #1: attached

Reviewer #2: Voltage imaging with genetically encoded indicators is a powerful emerging technique for measuring neural activity. In this manuscript, Cai et al. describe a suite of computational tools – VolPy – for extracting time series proportional to voltage changes from voltage imaging datasets. VolPy builds upon past algorithmic developments from these authors to provide a scalable pipeline for motion correction, ROI segementation, spike detection, and denoising. This tool efficiently handles large voltage imaging datasets and should be extremely useful for labs establishing voltage imaging as an experimental technique. The authors do a commendable job benchmarking VolPy on several existing voltage imaging datasets and demonstrate generalization to new datasets differing qualitatively from those used for training.

Major comments:

1) In the most likely use case, labs implementing VolPy will annotate additional training data and re-train the Mask R-CNN for best results. While the authors include a brief description of this process in the Discussion, the manuscript would benefit from some additional description of the tools included in VolPy for this purpose in the materials and methods and results. I suggest the authors include some quantification of how precision, recall, etc. change with increasing amounts of training data from new datasets differing qualitatively from those used for initial training. This would help give prospective users a better idea of the time investment that would be needed for adoption.

2) The manuscript appears to lack analysis of how spike identification is improved by the modified Spike Pursuit algorithm compared to other simpler approaches.

Minor comments:

Line 198: “More in details” appears to be a typo and should be omitted or otherwise corrected.

**Have all data underlying the figures and results presented in the manuscript been provided?**

Reviewer #1: Yes

Reviewer #2: Yes

PLOS authors have the option to publish the peer review history of their article (what does this mean?). If published, this will include your full peer review and any attached files.

Reviewer #1: No

Reviewer #2: No
---

## [Decision Letter · Decision Letter 1]

21 Jan 2021

Dear Dr. Giovannucci,

Thank you very much for submitting your manuscript "VolPy: automated and scalable analysis pipelines for voltage imaging datasets" for consideration at PLOS Computational Biology. As with all papers reviewed by the journal, your manuscript was reviewed by members of the editorial board and by several independent reviewers. The reviewers appreciated the attention to an important topic. Based on the reviews, we are likely to accept this manuscript for publication, providing that you modify the manuscript according to the review recommendations.

Sincerely,

Corey Acker

Guest Editor

PLOS Computational Biology

Kim Blackwell

Deputy Editor

PLOS Computational Biology

[LINK]

Reviewer's Responses to Questions

**Comments to the Authors:**

Reviewer #1: The authors have done an impressive job addressing my comments and questions, improving GT annotations and implementing comparisons to competing methods, highlighting both computational scalability, as well as better performance in extracting spatial footprints and spikes. I recommend the paper be accepted for publication.

Small comment regarding Figure 3 – using red and green contours is probably not an accessible choice for color-blind readers.

**Have all data underlying the figures and results presented in the manuscript been provided?**

Reviewer #1: Yes

PLOS authors have the option to publish the peer review history of their article (what does this mean?). If published, this will include your full peer review and any attached files.

Reviewer #1: No

Figure Files:

Data Requirements:

Reproducibility:

Suggested addition from the editor: Excellent job addressing the concerns of the reviewer and upgrading the paper and the tools. If possible, please add at least a discussion point to make sure readers are aware of fundamental differences between calcium and voltage imaging. Even though in practice it might not be that important, useful signal from voltage sensors are only from the cell membrane, not cytosol like most calcium sensors, and depending on details of the voltage sensor and imaging system (ex. resolution), this may or may not play an important role in how data are extracted/analyzed. I will leave the reviewer's point about figure line colors up to your discretion.

---

## [Editor Report · Decision Letter 2]

16 Feb 2021

Dear Dr. Giovannucci,

We are pleased to inform you that your manuscript 'VolPy: automated and scalable analysis pipelines for voltage imaging datasets' has been provisionally accepted for publication in PLOS Computational Biology.

Best regards,

Corey Acker

Guest Editor

PLOS Computational Biology

Kim Blackwell

Deputy Editor

PLOS Computational Biology

---

## [Editor Report · Acceptance letter]

8 Apr 2021

PCOMPBIOL-D-20-00666R2 

VolPy: automated and scalable analysis pipelines for voltage imaging datasets

Dear Dr Giovannucci,

I am pleased to inform you that your manuscript has been formally accepted for publication in PLOS Computational Biology. Your manuscript is now with our production department and you will be notified of the publication date in due course.

With kind regards,

Katalin Szabo
